# OS-PCA: Orthogonal Smoothed Principal Component Analysis Applied to Metabolome Data

**DOI:** 10.3390/metabo11030149

**Published:** 2021-03-05

**Authors:** Hiroyuki Yamamoto, Yasumune Nakayama, Hiroshi Tsugawa

**Affiliations:** 1Human Metabolome Technologies, Inc., 246-2 Mizukami, Kakuganji, Tsuruoka, Yamagata 997-0052, Japan; 2Department of Applied Microbial Technology, Sojo University, 4-22-1 Ikeda, Kumamoto 860-0082, Japan; ynaka@bio.sojo-u.ac.jp; 3RIKEN Center for Sustainable Resource Science, 1-7-22 Suehiro-cho, Tsurumi-ku, Yokohama, Kanagawa 230-0045, Japan; hiroshi.tsugawa@riken.jp; 4RIKEN Center for Integrative Medical Sciences, 1-7-22 Suehiro-cho, Tsurumi-ku, Yokohama, Kanagawa 230-0045, Japan; 5Graduate School of Medical Life Science, Yokohama City University, 1-7-22 Suehiro-cho, Tsurumi-ku, Yokohama, Kanagawa 230-0045, Japan

**Keywords:** principal component analysis, smoothing, statistical hypothesis testing, metabolomics

## Abstract

Principal component analysis (PCA) has been widely used in metabolomics. However, it is not always possible to detect phenotype-associated principal component (PC) scores. Previously, we proposed a smoothed PCA for samples acquired with a time course or rank order, but hypothesis testing to select significant metabolite candidates was not possible. Here, we modified the smoothed PCA as an orthogonal smoothed PCA (OS-PCA) so that statistical hypothesis testing in OS-PC loadings could be performed with the same PC projections provided by the smoothed PCA. Statistical hypothesis testing is especially useful in metabolomics because biological interpretations are made based on statistically significant metabolites. We applied the OS-PCA method to two real metabolome datasets, one for metabolic turnover analysis and the other for evaluating the taste of Japanese green tea. The OS-PCA successfully extracted similar PC scores as the smoothed PCA; these scores reflected the expected phenotypes. The significant metabolites that were selected using statistical hypothesis testing of OS-PC loading facilitated biological interpretations that were consistent with the results of our previous study. Our results suggest that OS-PCA combined with statistical hypothesis testing of OS-PC loading is a useful method for the analysis of metabolome data.

## 1. Introduction

Principal component analysis (PCA) [1] and partial least squares (PLS) [2,3] have been widely applied to metabolome data [4]. PCA is an unsupervised method that does not require group information for its computation, whereas PLS has been used as a supervised method. Various multivariate analysis methods that utilize additional information also have been proposed to analyze metabolome data [5]. These methods make it possible to extract features that are especially suited to a specific purpose. Smilde et al. [6] reviewed multivariate analysis methods for time-resolved and longitudinal metabolome data (which they called dynamic metabolomic data analysis), such as smooth-PCA, which combines PCA with smoothness, and analysis of variance (ANOVA) simultaneous component analysis (ASCA) [7,8]. Dynamic PCA was extended by including a probabilistic model—probabilistic dynamic PCA [9]. We previously combined PCA, PLS, and Fisher discriminant analysis (FDA) with smoothness as smoothed PCA, smoothed PLS, and smoothed FDA, and the kernel-based nonlinear extension for this type of data [10]. The smoothed PCA can extract features that are associated with phenotypes such as time course information in the principal component (PC) scores.

A typical PCA of metabolome data usually involves three steps [11]. In the first step, the samples are visualized using PC scores and PCs that are associated with phenotypes such as time course information are found. In the second step, significant metabolites are selected by loadings defined by the eigenvector, univariate analysis such as t-test, or often manual inspection. In the third step, the associations between the significant metabolites and metabolic pathways are analyzed. In the second step, there are many possible univariate analysis approaches [12,13] that can be used to select significant metabolites. Metabolite selection using loadings in ordinary multivariate analysis such as PCA has the disadvantage that meaningful metabolites cannot be selected when no interpretable feature is available, such as time course information in PC scores. Compared with supervised multivariate analysis such as PLS, in unsupervised multivariate analysis, metabolite selection using loadings is not limited to group differences when interpretable features can be found in PC scores. To extract interpretable features, it is preferable to apply methods that are suitable for real metabolome data, e.g., smoothed PCA for time course data.

Another issue in metabolite selection using loadings is that metabolites are often selected subjectively (e.g., the top 10 metabolites), which can lead to biased biological inferences because these inferences are made for metabolites that are not always statistically significant. In PCA, this is not a major problem because PC loadings can be considered as correlation coefficients between PC scores and the level of each metabolite when the level is scaled to unit variance. This characteristic can be used to select metabolites by statistical hypothesis testing of PC loadings in PCA [11]. PLS and its extension with the rank order of groups (PLS-ROG) also can be used to select statistically significant metabolites by loadings [14]. In our previous formulation of smoothed PCA, it was difficult to explain the statistical properties of loadings defined by eigenvectors, so statistically significant metabolites could not be selected using loadings such as ordinary PCA and PLS.

In this study, we describe an orthogonal smoothed PCA (OS-PCA) method that was designed to handle the same type of data that smoothed PCA deals with, for example, samples that were acquired with a time course or rank order. OS-PCA can resolve the issues about smoothed PCA loadings because OS-PC loadings can be interpreted as the correlation coefficient between OS-PC scores of an auxiliary variable and the level of each metabolite. Therefore, significant metabolites can be selected by statistical hypothesis testing of the loadings in OS-PCA as well as in PCA. Additionally, the formulation of OS-PCA is simpler than that of smoothed PCA because smoothed PCA is formulated as a generalized eigenvalue problem whereas OS-PCA is formulated as an eigenvalue problem. OS-PCA also has the advantage that the core part of the computation can be implemented using a few lines of programming. We applied OS-PCA to two real metabolome datasets, one for metabolic turnover analysis [15] and the other to evaluate the taste of Japanese green tea [16,17]. All the computations in this study were performed using R software and the programs are freely available on our website (https://github.com/hiroyukiyamamoto/os-pca, accessed on 4 March 2021).

## 2. Results and Discussion

We applied the OS-PCA method to two real metabolome datasets to verify its usefulness.

### 2.1. Case Study 1: Metabolic Turnover Analysis

In this case study, we used the metabolic turnover data reported by Nakayama et al. [15]. This dataset contains data for three groups: *Saccharomyces cerevisiae* BY4742 cultured in sucrose-dextran (SD) medium with amino acid supplement and *S. cerevisiae* X2180 cultured in SD medium with or without amino acid supplement. In all three groups, the culture medium contained ^13^C-labeled glucose. The culture fluid was sampled at 0, 10, 20, 40, 80, 160, 320, 640, 1280, 2560, and 7200 s. Metabolome data measured by gas chromatography/mass spectrometry were converted to isotopomer ratios, i.e., the ratio of peak area of metabolites (amino acids) labeled with the ^13^C isotope to peak area of metabolites with nonisotopic ^12^C.

We performed a PCA for autoscaled data and confirmed the relation of incubation time to metabolic turnover in PC1 (Figure 1). In PC1, the sample order of the score was consistent with that of the incubation time (Figure 1b), which suggests that time course information can be extracted in PC1. However, differences between groups, such as strains and culture mediums, were not detected in PC1 and PC2. The contribution ratios of PC1 and PC2 were 65.96% and 9.596%, respectively, so the cumulative contribution ratio was over 75% for PC1 and PC2. We also calculated PC3, PC4, and PC5 scores (Appendix A). We confirmed the fluctuation trend against incubation time of X2180 cultured in SD medium without amino acids differed from the trends of BY4742 and X2180 cultured in SD medium with amino acids in PC4 and PC5. However, the contribution ratios of PC4 and PC5 were very small—3.80% and 0.88%, respectively—so we did not consider these PCs in the subsequent analysis.

Nakayama et al. [15] applied an ad-hoc transformation to this same data, whereby the average value was subtracted from every incubation time to show the differences between groups. As a result, they could confirm group differences in PC1 but not time course information [15].

We also performed smoothed PCA and OS-PCA (Figure 2) for this data with only autoscaling and no ad-hoc transformation. We subjectively set the smoothing parameter *κ* to 0.1 in the smoothed PCA and to 0.999 (i.e., close to 1) in the OS-PCA, and used the second differential matrix **D^(2)^** (see Section 3.1 for details). 

The smoothed PC (Figure 2a) and OS-PC scores (Figure 2b) were almost the same, although the positive and negative directions were reversed. This result showed that OS-PCA was able to extract the same features as smoothed PCA and confirmed the PCA result as well as the relation of incubation time to metabolic turnover in smoothed PC1 (Figure 2a) and OS-PC1 (Figure 2b). However, unlike the PCA result of PC1 and PC2, the fluctuation trend against the incubation time of X2180 cultured in SD medium without amino acids differed from the trends of BY4742 and X2180 cultured in SD medium with amino acids in smoothed PC2 (Figure 2a) and OS-PC2 (Figure 2b). The different trends of strains and culture mediums were clear in the OS-PC2 scores of auxiliary variables (Figure 2c).

Together these results show that the PCA extracted only the time course information, but not group differences, within PC1 and PC2, and after ad-hoc transformation [15], the PCA extracted group differences, but not time course information. Smoothed PCA and OS-PCA successfully extracted both time course information and group differences as a major component. These results indicate the usefulness of smoothed PCA and OS-PCA applied to metabolome data with a time course and groups.

Next, we selected statistically significant correlated metabolites using OS-PC2 loading. Lysine_3TMS_Minor (R = 0.5109, *p* = 0.0024, *q* = 0.0357), Lysine_4TMS_Major (R = 0.5207, *p* = 0.0019, *q* = 0.0357), Histidine (R = 0.7110, *p* = 3.533 × 10^−6^, *q* = 0.0001), and Peak-63 (R = 0.7142, *p* = 3.045 × 10^−6^, *q* = 0.0001) levels showed statistically significant positive correlations (*q* < 0.05) with OS-PC2 scores (Appendix A). As described in Section 3.4, the correlation coefficient between the OS-PC score of auxiliary variables and each metabolite level was defined as OS-PC loading. Therefore, these metabolite levels are highly correlated with the OS-PC2 scores of the auxiliary variables (Figure 2c), but not always correlated with the OS-PC2 scores (Figure 2b).

Nakayama et al. [15] identified Peak-63 as a histidine derivative by narrowing down candidates using the hypothesis that the similarity of isotopomer ratios corresponded to distances on the metabolic pathway map. In the OS-PCA, Peak-63 had the highest OS-PC2 loading followed by Histidine with the second highest. No other statistically significant unidentified peaks were found. These results support those of Nakayama et al., who identified Peak-63 as a histidine derivative. We found that the OS-PC2 score decreased with incubation time only for X2180 cultured in SD medium with amino acids. Because Histidine and Lysine showed statistically significant positive correlations with the OS-PC2 score, we selected Histidine and Lysine as the metabolites that decreased with incubation time only for X2180 cultured in SD medium with amino acids. Because these metabolites were not labeled with the ^13^C isotope, Nakayama et al. [15] concluded that they were not synthesized under this condition. Branched-chain amino acids and intermediates of the tricarboxylic acid (TCA) cycle, Isoleucine_2 trimethylsilyl (TMS) and Citric acid + Isocitric acid, were among the top 10 metabolites in OS-PC2 loading, but this result was not statistically significant. This may be because, although the difference in the fluctuation trend against incubation time was shown in OS-PC2, the group separation was not shown clearly.

The metabolites that we focused on in the OS-PCA were partially consistent with those of the previous study [15]. The OS-PCA and smoothed PCA both detected differences in the fluctuation trend with incubation time, but the OS-PCA did not completely separate the groups. This may be because OS-PCA and smoothed PCA are unsupervised, not supervised, methods. To separate groups more clearly, a supervised approach such as PLS also could be used.

### 2.2. Case Study 2: Metabolome Analysis for the Taste of Japanese Green Tea

In this case study, we used some of the metabolome data from the Platform for RIKEN Metabolomics (http://prime.psc.riken.jp/Metabolomics_Software/StatisticalAnalysisOnMicrosoftExcel/index.html, accessed on 4 March 2021) to evaluate the taste of Japanese green tea [16,17]. In the selected dataset, each green tea leaf that ranked 1, 6, 11, 16, 21, 31, 36, 41, 46, and 51 in tasting was measured three times.

We performed a PCA (Figure 3) and confirmed that the differences among the green tea leaf samples were reflected in PC1 and the partial association with the taste ranking was reflected in PC2; however, no clear association with the taste ranking was detected. We also calculated PC3, PC4, and PC5 scores (Appendix A), but their associations with the taste ranking were not confirmed. Then, we applied the OS-PCA for repeated measurement data (*κ* = 0.1) to the same data (Figure 4). As described in Section 3.3, OS-PC scores for repeated measurement data were calculated by **t** = **Xw_x_** (Figure 4a) as was done in case study 1, whereas OS-PC scores for repeated data of auxiliary variables were calculated by **Ms** = **MXw_y_** (Figure 4b), which is the score for the average of repeated measures. Therefore, the OS-PC scores of auxiliary variables can be regarded as the same value for repeated data.

We calculated the correlation coefficient between the OS-PC1 score and the taste ranking to confirm the effect of the smoothing parameter *κ* (Appendix A). For this, we set *κ* = 0.1 because the correlation coefficient did not change much for *κ* = 0.1 to 0.999, and used the second differential matrix **D^(2)^** (see Section 3.1 for details).

The OS-PC1 score roughly reflected the taste ranking (Figure 4a), which suggests that the metabolome data included metabolites that reflected the taste ranking. Furthermore, the sample order of the OS-PC1 score of auxiliary variables (Figure 4b) was completely consistent with the samples’ taste ranking. Sample order of OS-PC1 score of auxiliary variables that is consistent with the correct order (e.g., taste ranking) is important because OS-PC loading is the correlation coefficient between the OS-PC score of auxiliary variables and each metabolite level, as described in Section 3.4.

To select statistically significant metabolites, statistical hypothesis testing of OS-PC1 loading was performed (Appendix A). Among the 225 detected metabolites, 32 (*p* < 0.05) and 14 (*q* < 0.25) were statistically significant. Among the 14 metabolites with *q* < 0.25, those with the highest OS-PC1 loading values (|R| > 0.7) were Raffinose (R = −0.9445, *p* = 3.888 × 10^−5^, *q* = 8.747 × 10^−3^), Adipic acid (R = −0.8311, *p* = 2.891 × 10^−3^, *q* = 0.1504), threo-3-Hydroxy-l-aspartic acid (R = −0.8158, *p* = 4.009 × 10^−3^, *q* = 0.1504), Arabinose (R = −0.8114, *p* = 4.381 × 10^−3^, *q* = 0.1504), Serine (R = 0.7940, *p* = 6.088 × 10^−3^, *q* = 0.1504), Shikimic acid (R = −0.7891, *p* = 6.655 × 10^−5^, *q* = 0.1504), and Galactose (R = −0.7660, *p* = 9.783 × 10^−3^, *q* = 0.1839).

In a previous study [16], quinic acid, amino acids, and sugars were associated with the taste ranking. The sugars raffinose, arabinose, and galactose showed statistically significant negative correlation with OS-PC1 scores (*q* < 0.25), which indicated these metabolites were present at high levels in the highly ranked teas. Among the amino acids, serine showed a statistically significant positive correlation with OS-PC1 scores (*q* < 0.25), which indicated this metabolite was present at a low level in the highly ranked teas. The statistical significance of quinic acid in green tea leaf was not confirmed in the OS-PCA.

## 3. Methods

### 3.1. Smoothed Principal Component Analysis (PCA)

In a previous study [10], we proposed a smoothed PCA method in which a smoothing term [18] was added to the constraint condition of PCA. The smoothed PCA is formulated as:(1)$$\begin{array}{c} \max\mathrm{var}\left(t\right)=\left(1/n\right){w_{x}}^{\prime }x^{\prime }Xw_{x}\\ \mathrm{subject}\;\mathrm{to}\;\left(1-\kappa \right){w_{x}}^{\prime }w_{x}+\kappa {\left(Dt\right)}^{\prime }\left(Dt\right)=1’ \end{array}$$
where **X** is a mean-centered data matrix with a sample in each row and metabolites in each column; **w_x_** is a weight vector; var(**t**) indicates variance of PC score vector **t** = **Xw_x_**, which is a linear combination of each variables in data matrix **X**; *n* is the number of samples; and *κ* is a smoothing parameter. **D** is the first or second differential matrix that is set as:$$D^{\left(1\right)}=\left[\begin{array}{cccccc} 1 & -1 & 0 & \cdots & 0 & \;0\\ 0 & 1 & -1 & \cdots & 0 & 0\\ \vdots & \vdots & \vdots & \ddots & \vdots & \vdots \\ 0 & 0 & 0 & \cdots & 1 & -1 \end{array}\right]\;D^{\left(2\right)}=\left[\begin{array}{cccccccc} 1 & -2 & 1 & 0 & \cdots & 0 & 0 & 0\\ 0 & 1 & -2 & -1 & \cdots & 0 & 0 & 0\\ \vdots & \vdots & \vdots & \vdots & \ddots & \vdots & \vdots & \vdots \\ 0 & 0 & 0 & 0 & \cdots & 1 & -2 & 1 \end{array}\right]$$

The first differential matrix **D**^(**1**)^ is (n − g) × n and second differential matrix **D**^(**2**)^ is (n − 2g) × n, where n is the number of samples and g is the number of groups. Finally, the smoothed PCA is written as a generalized eigenvalue problem:(2)$$\left(1/n\right)X^{\prime }Xw_{x}=\lambda \left\{\left(1-\kappa \right)I+\kappa X^{\prime }D^{\prime }\mathrm{DX}\right\}w_{x}$$
where *λ* is an eigenvalue and **I** is the identity matrix. When *κ* is set to 0, the smoothed PCA is consistent with the PCA. Theoretical details of the smoothed PCA are given in [10]. In the smoothed PCA, it is difficult to explain the eigenvector **w_x_** statistically, so statistically significant metabolites cannot be selected using loadings defined by the eigenvector.

### 3.2. Orthogonal Smoothed Principal Component Analysis (OS-PCA)

We introduced an auxiliary variable **s** (= **Xw_y_**) and maximized the covariance between **t** and **s** instead of the variance of **t** in PCA as:(3)$$\begin{array}{c} \max\mathrm{cov}\left(t,s\right)=\left(1/n\right){w_{x}}^{\prime }x^{\prime }Xw_{y}\\ s.t.\;{w_{x}}^{\prime }w_{x}=1,\;\left(1-\kappa \right){w_{y}}^{\prime }w_{y}+\kappa {\left(Ds\right)}^{\prime }\left(Ds\right)={w_{y}}^{\prime }Pw_{y}=1 \end{array}$$
where **w_y_** is a weight vector of auxiliary variable; matrix **P** is (1 − *κ*)**I** + *κ***X′D′DX**; and cov(**t**,**s**) indicates covariance between score vectors **t** and **s**. This formulation is similar to that of PLS-ROG [14]. The main difference between smoothed PCA and PLS-ROG is that smoothed PCA does not use a response variable, such as group information. In PLS and PLS-ROG, the response variable has an important role when loadings are interpreted statistically. Similarly, the auxiliary variable **s** of OS-PCA is essential to interpret OS-PC loading statistically so that statistical hypothesis testing of OS-PC loadings can be performed, as explained in Section 3.4.

Using the Lagrange multipliers method, Equation (3) was reformulated as the maximization of:(4)$$J=\left(1/n\right)w_{x}^{\prime }X^{\prime }Xw_{y}+\lambda_{x}\left(1-{w_{x}}^{\prime }w_{x}\right)+\lambda_{y}\left(1-{w_{y}}^{\prime }Pw_{y}\right),$$
where λ_x_ and λ_y_ are Lagrange multipliers. Partial differentiation of Equation (4) with respect to **w**_x_ and **w**_y_ followed by a transformation, yields two equations:(5)$$\begin{array}{c} \left(1/n\right)X^{\prime }Xw_{y}=2\lambda_{x}w_{x}\\ \left(1/n\right)X^{\prime }Xw_{x}=2\lambda_{y}Pw_{y}\; \end{array}.$$

So that both of these equations can express the eigenvalue problem for **w_x_** and **w_y_**, we rearranged them as:(6)$$\begin{array}{c} \left(1/n^{2}\right)X^{\prime }XP^{-1}X^{\prime }Xw_{x}=\lambda w_{x}\\ \left(1/n^{2}\right)X^{\prime }XX^{\prime }Xw_{y}=\lambda Pw_{y} \end{array}$$
where λ = 4λ_x_λ_y_. Like PCA, OS-PCA has a unique solution because it was formulated as an eigenvalue problem for **w_x_**, and the i-th weight vector of OS-PCA corresponds to the i-th largest eigenvector in Equation (6). The smoothed PCA proposed previously [10] was formulated as a generalized eigenvalue problem, where the eigenvectors were not orthogonal with each other. For the OS-PCA, the formulation was written as an eigenvalue problem for **w_x_**, and the eigenvectors are orthogonal with each other. When κ is set to 0, the matrix **P** becomes the identity matrix and the two equations of eigenvalue problems for **w_x_** and **w_y_** in Equation (6) are the same. Therefore, **w_x_** and **w_y_** are the same eigenvector, which corresponds to a specific eigenvalue. When **w_x_** and **w_y_** are the same, the two equations of the eigenvalue problems in Equation (5) are the same and consistent with ordinary PCA.

The contribution ratio of PCA is associated with the variance of PC scores because it corresponds to the ratio of variance of a specific PC score to the sum of variance of all PCs. Conversely, the contribution ratio of OS-PCA is associated with the covariance between OS-PC scores of explanatory and auxiliary variables. Because the contribution ratios of PCA and OS-PCA are associated with different statistics, they cannot be compared.

### 3.3. OS-PCA for Repeated Measurement Data

Both smoothed PCA and OS-PCA assume that all the samples are ordered. However, one sample may be measured repeatedly to reduce the effect of variability of measurement. In such a case, ordered and unordered samples of repeated measurements will be mixed in the data, so the simplest and most straightforward method for the OS-PCA is to use the averaged data for the repeated measurement. Alternatively, the averaging operation can be combined with the OS-PCA for repeated measurement data and formulated as:(7)$$\begin{array}{c} \max\mathrm{cov}\left(Mt,Ms\right)=\left(1/n\right){w_{x}}^{\prime }X^{\prime }M^{\prime }MXw_{y}\\ s.t.\;{w_{x}}^{\prime }w_{x}=1,\;\left(1-\kappa \right){w_{y}}^{\prime }w_{y}+\kappa {\left(DMs\right)}^{\prime }\left(DMs\right)={w_{y}}^{\prime }Qw_{y}=1, \end{array}$$
which is similar to the formulation in Section 3.2. The averaging matrix **M** is set as:$$M=\left[\begin{array}{ccccc} m_{1} & 0 & 0 & \cdots & 0\\ 0 & m_{2} & 0 & \cdots & 0\\ 0 & 0 & m_{3} & \cdots & 0\\ \vdots & \vdots & \vdots & \ddots & \vdots \\ 0 & 0 & 0 & \cdots & m_{n} \end{array}\right]$$

The vector is **m_1_** = [1/n_1_, 1/n_1_, 1/n_1_, …, 1/n_1_] and each element is the reciprocal of the number of measurement repetitions. Similar to the formulation used in Section 3.2, Equation (7) can be reformulated using the Lagrange multipliers method as:(8)$$J=\left(1/n\right)w_{x}^{\prime }X^{\prime }M^{\prime }MXw_{y}+\lambda_{x}\left(1-{w_{x}}^{\prime }w_{x}\right)+\lambda_{y}\left(1-{w_{y}}^{\prime }Qw_{y}\right),$$
where matrix **Q** is (1 − *κ*)**I** + *κ***X′M′D′DMX**.

Partial differentiation of Equation (8) with respect to **w_x_** and **w_y_** yields two equations:(9)$$\begin{array}{c} \left(1/n\right)X^{\prime }M^{\prime }MXw_{y}=2\lambda_{x}w_{x}\\ \left(1/n\right)X^{\prime }M^{\prime }MXw_{x}=2\lambda_{y}Qw_{y} \end{array}.$$

These equations are rearranged to express the eigenvalue problem for **w_x_** and **w_y_** as:(10)$$\begin{array}{c} \left(1/n^{2}\right)X^{\prime }M^{\prime }MXQ^{-1}X^{\prime }M^{\prime }MXw_{x}=\lambda w_{x}\\ \left(1/n^{2}\right)X^{\prime }M^{\prime }MXX^{\prime }M^{\prime }MXw_{y}=\lambda Qw_{y} \end{array}.$$

As we did for the OS-PCA in Section 3.2, the formulation of OS-PCA for repeated data was written as an eigenvalue problem for **w_x_**. OS-PC scores for repeated measurement data were calculated by **t** = **Xw_x_**, whereas OS-PC scores for repeated measurement data of auxiliary variables were calculated by **Ms** = **MXw_y_**, which is the score for the average of repeated measures. Therefore, the OS-PC scores of auxiliary variables can be regarded as the same value for repeated measurement data.

In our previous smoothed PCA [10], we did not consider repeated measurement of the same sample. In the OS-PCA, we applied smoothing for the averaged value of repeated measurements, and the OS-PCA result for repeated measurement data did not affect the order of samples within the repeated measurements.

### 3.4. Statistical Property of OS-PC Loading for Autoscaled Data

To select metabolites using statistical criteria, it is essential to clarify the statistical property of **w_x_**. The correlation coefficient between **s** and **x_p_**, the p-th variable of data matrix **X**, is written as:(11)$$\mathrm{corr}\left(s,x_{p}\right)=\mathrm{cov}\left(s,x_{p}\right)/\sqrt{\mathrm{var}\left(s\right)}\sqrt{\mathrm{var}\left(x_{p}\right)},$$
where corr(**s**,**x_p_**) indicates the correlation coefficient between the OS-PC score vectors **s** and **x_p_**. When data matrix **X** is scaled to zero mean and unit variance for each variable (i.e., autoscaling), Equation (11) is written as:(12)$$\mathrm{corr}\left(s,x_{p}\right)=\mathrm{cov}\left(s,x_{p}\right)/\sqrt{\mathrm{var}\left(s\right)}$$
because variance of **x_p_** is 1. Then, **s** = **Xw_y_** is substituted into Equation (12) as:(13)$$\begin{array}{c} \mathrm{corr}\left(s,x_{p}\right)=\left(1/n\right)w_{y}^{\prime }X^{\prime }Xc/\sqrt{\left(1/n\right){w_{y}}^{\prime }X^{\prime }Xw_{y}}\\ =c^{\prime }\left\{\left(1/n\right)X^{\prime }Xw_{y}\right\}/\sqrt{\left(1/n\right){w_{y}}^{\prime }X^{\prime }Xw_{y}} \end{array},$$
where **c** is introduced as the column vector in which the p-th element is 1 and the other elements are 0, giving **x_p_** = **Xc**. Then, $\left(1/n\right)X^{\prime }Xw_{y}=2\lambda_{x}w_{x}$ is substituted into Equation (13) as:(14)$$\mathrm{corr}\left(s,x_{p}\right)=c^{\prime }\left(2\lambda_{x}w_{x}\right)/\sqrt{\left(1/n\right){w_{y}}^{\prime }X^{\prime }Xw_{y}}=2\lambda_{x}w_{x,p}/\sqrt{\left(1/n\right){w_{y}}^{\prime }X^{\prime }Xw_{y}}.$$

The denominator of Equation (14) is not affected by the p-th variable. Therefore, the eigenvector **w_x_** is proportional to the correlation coefficient between **s** and **x_p_**. We defined this statistic of the correlation coefficient between the OS-PC score of auxiliary variables and each metabolite level as OS-PC loading. We set **r** as corr(**s**,**x_p_**) and performed statistical hypothesis testing of the correlation coefficient using a t-statistic as:(15)$$t-\mathrm{statistic}=r\sqrt{n-2}/\sqrt{1-r^{2}},$$
which has a t-distribution with *n* − 2 degrees of freedom [11,14]. The result is used to select significant metabolites by statistical hypothesis testing of OS-PC loading in OS-PCA as well as PCA.

### 3.5. Statistical Property of OS-PC Loading for Repeated Measurements and Autoscaled Data

The correlation coefficient between averaged score **Ms** and averaged p-th metabolite levels **Mx_p_** is written as:(16)$$\mathrm{corr}\left(Ms,Mx_{p}\right)=\mathrm{cov}\left(Ms,Mx_{p}\right)/\sqrt{\mathrm{var}\left(Ms\right)}\sqrt{\mathrm{var}\left(Mx_{p}\right)}$$

In OS-PCA, the data matrix **X** is scaled by autoscaling for each metabolite level, whereas the averaged data matrix **MX** is transformed using autoscaling in OS-PCA for repeated measurement data. This means that the averaged data for repeated measurements are transformed by autoscaling. Then, Equation (16) is written as:(17)$$\mathrm{corr}\left(Ms,Mx_{p}\right)=\mathrm{cov}\left(Ms,Mx_{p}\right)/\sqrt{\mathrm{var}\left(Ms\right),}$$

**s** = **Xw_y_** is substituted into Equation (17) as:(18)$$\begin{array}{c} \mathrm{corr}\left(Ms,Mx_{p}\right)=\left(1/n\right)w_{y}^{\prime }X^{\prime }MX^{\prime }MXc/\sqrt{\left(1/n\right){w_{y}}^{\prime }X^{\prime }M^{\prime }MXw_{y}}\\ =c^{\prime }\left\{\left(1/n\right)X^{\prime }M^{\prime }MXw_{y}\right\}/\sqrt{\left(1/n\right){w_{y}}^{\prime }X^{\prime }M^{\prime }MXw_{y}}, \end{array}$$
and (1 − *n*)**X′M′MXw_y_** = 2λ_x_**w_x_** is substituted into Equation (18) as:(19)$$\begin{array}{c} \mathrm{corr}\left(Ms,Mx_{p}\right)=c^{\prime }\left(2\lambda_{x}w_{X}\right)/\sqrt{\left(1/n\right)w_{y}^{\prime }X^{\prime }M^{\prime }MXw_{y}}\\ =2\lambda_{x}w_{x,p}/\sqrt{\left(1/n\right)w_{y}^{\prime }X^{\prime }M^{\prime }MXw_{y}}. \end{array}$$

The denominator of Equation (19) is not affected by the p-th variable. Therefore, the eigenvector **w_x_** is proportional to the correlation coefficient between averaged score **Ms** and the averaged level of each metabolite **Mx_p_** for repeated measurement data, so statistical hypothesis testing of OS-PC loading for repeated measurement data can be performed in the same way as OS-PCA. The main features of the PCA, smoothed PCA, and OS-PCA methods are summarized in Table 1.

## 4. Conclusions

We developed OS-PCA as an improved version of the smoothed PCA. OS-PCA can be used to perform statistical hypothesis testing of OS-PC loading, which is very important in metabolomics because biological interpretations are made on the basis of the determined significant metabolites. We applied OS-PCA to two metabolomics datasets, which confirmed the usefulness of OS-PCA in detecting significant metabolites. We expect that OS-PCA can be usefully applied to metabolome data that have external information such as time course and rank order of samples.

## Figures and Tables

**Figure 1 metabolites-11-00149-f001:**
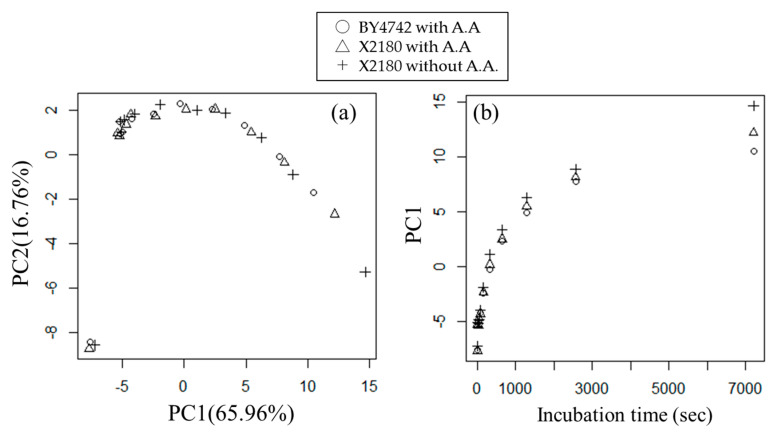
Scatter plot of PC scores obtained by PCA of the metabolic turnover data of Nakayama et al. [15]. (**a**) Scatter plot of first and second PC scores (PC1 and PC2). The contribution ratios (variance) of PC1 and PC2 were 65.96% and 16.76%, respectively. (**b**) Scatter plot of PC1 score and incubation time. (○) *S. cerevisiae* BY4742 cultured in SD medium with amino acids (A.A.), (Δ) *S. cerevisiae* X2180 cultured in SD medium with amino acids, (+) *S. cerevisiae* X2180 cultured in SD medium without amino acids.

**Figure 2 metabolites-11-00149-f002:**
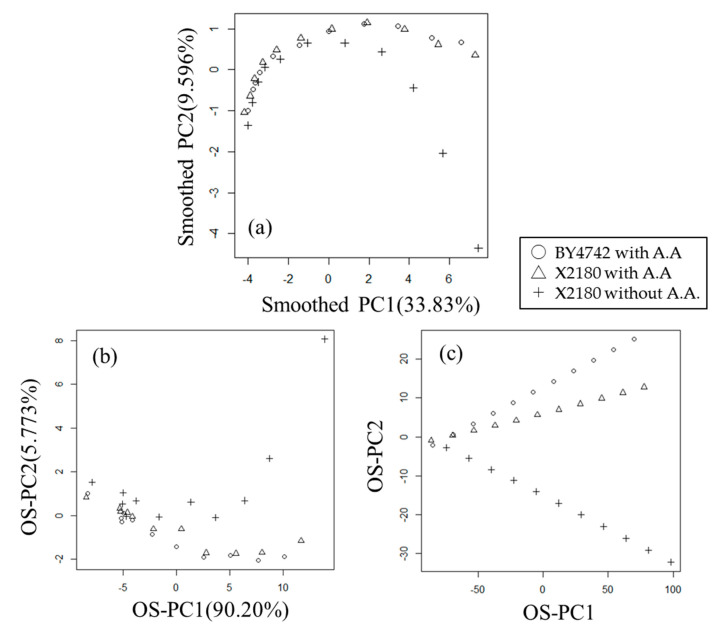
Scatter plots of first and second PC scores obtained by smoothed PCA and OS-PCA of the metabolic turnover data of Nakayama et al. [15]. (**a**) Scatter plot of first and second smoothed PC scores (PC1 and PC2) obtained by smoothed PCA (*κ* = 0.1) with the second differential matrix D^(2)^. The contribution ratios (variance) of smoothed PC1 and smoothed PC2 were 33.83% and 9.596%, respectively. (**b**) Scatter plot of first and second OS-PC scores (OS-PC1 and OS-PC2) obtained by OS-PCA (*κ* = 0.999) with the second differential matrix D^(2)^. The contribution ratios (covariance) of OS-PC1 and OS-PC2 were 90.20% and 5.773%, respectively. (**c**) Scatter plot of OS-PC scores of auxiliary variables. (○) *S. cerevisiae* BY4742 cultured in SD medium with amino acids (A.A.), (Δ) *S. cerevisiae* X2180 cultured in SD medium with amino acids, (+) *S. cerevisiae* X2180 cultured in SD medium without amino acids.

**Figure 3 metabolites-11-00149-f003:**
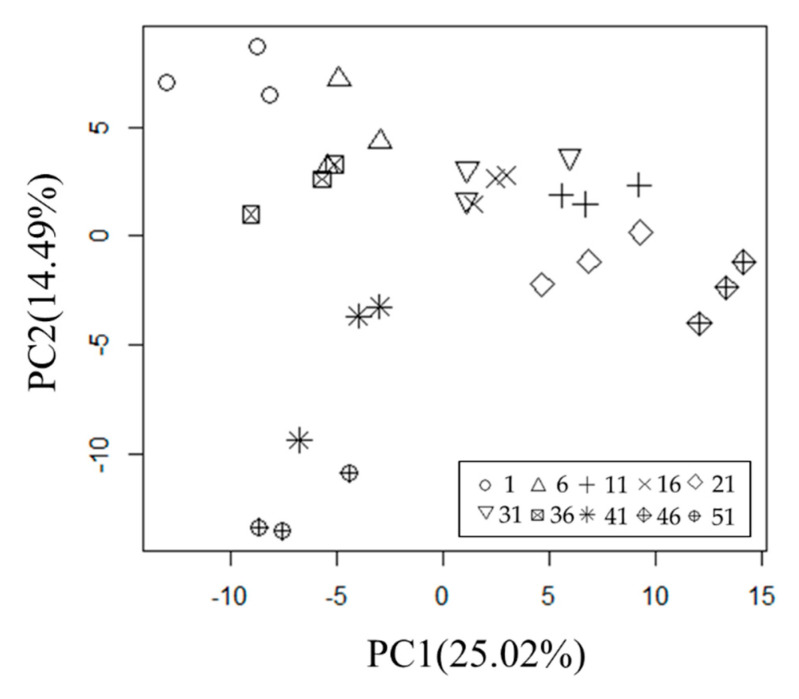
Scatter plot of first and second PC scores (PC1 and PC2) obtained by PCA of metabolome data for taste testing of Japanese green tea. The contribution ratios (variance) of PC1 and PC2 were 25.02% and 14.49%, respectively. The tea leaf ranks were (○) 1, (Δ) 6, (+) 11, (×) 16, (◇) 21, (∇) 31, (⊠) 36, (🞽) 41, (
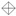
) 46, (⊕) 51.

**Figure 4 metabolites-11-00149-f004:**
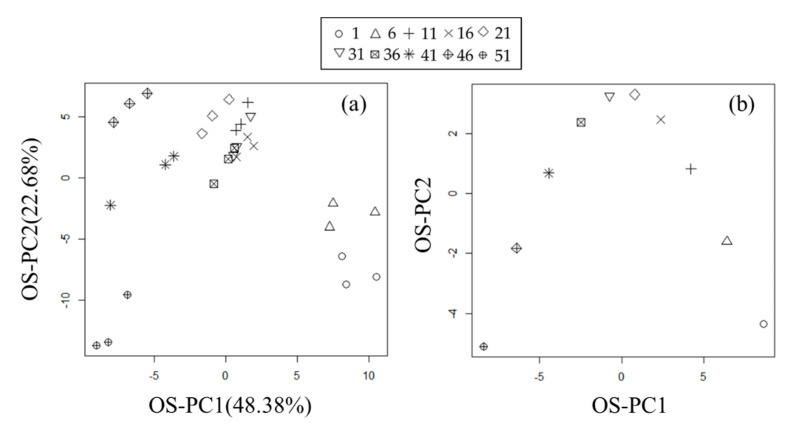
Scatter plots of first and second OS-PC scores (OS-PC1 and OS-PC2) obtained by OS-PCA (*κ* = 0.1) of the metabolome data for taste testing of Japanese green tea. (**a**) Scatter plot of OS-PC scores. The contribution ratios (covariance) of OS-PC1 and OS-PC2 were 55.88% and 21.03%, respectively. (**b**) Scatter plot of OS-PC scores of auxiliary variables for the average of repeated measures. The tea leaf ranks were (○) 1, (Δ) 6, (+) 11, (×) 16, (◇) 21, (∇) 31, (⊠) 36, (🞽) 41 (
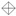
) 46, (⊕) 51.

**Table 1 metabolites-11-00149-t001:** Main features of the PCA, smoothed PCA, and OS-PCA methods.

Method	Equation	Eigenvector	Hypothesis Testing
PCA	(1/*n*)**X′Xw_x_** = *λ***w_x_**	**w_x_**∝corr(**t**,**x_p_**)	Applicable
Smoothed PCA	(1/*n*)**X′Xw_x_** = *λ*{(1 − *κ*)**I** + *κ***X′D′DX**}**w_x_**	Not Available	Not Applicable
OS-PCA	(1/*n*^2^)**X′XP**^−1^**X′Xw_x_** = *λ***w_x_**(1/*n*^2^)**X′XX′Xw_y_** = *λ***Pw_y_**	**w_x_**∝corr(**s**,**x_p_**)	Applicable

**X**: data matrix; **w_x_**: weight vector; **w_y_**: weight vector of auxiliary variable; n: number of samples; *λ*: eigenvalue; κ: smoothing parameter; **P**: (1 − *κ*)**I** + *κ***X′D′DX**; **I**: identity matrix; **D**: differential matrix; **t**: score vector (**t** = **Xw_x_**); **s**: score vector of auxiliary variable (**s** = **Xw_y_**); **x_p_**: p-th variable.

## Data Availability

The data presented in this study are openly available from this link: https://github.com/hiroyukiyamamoto/os-pca.

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
