# Peer review of "OS-PCA: Orthogonal Smoothed Principal Component Analysis Applied to Metabolome Data"

_metabolites, 2021, doi:10.3390/metabo11030149_

Round 1

Reviewer 1 Report

The authors present orthogonal smoothed PCA as a method for analyzing time course or repeated measures data, as illustrated with two metabolomics data sets. It is an interesting addition to existing tools for analyzing these types of data.

The methods section is rather technical and might not appeal to everybody. The uninitiated reader could gain a lot from a description in words how smoothed PCA differs from PCA and what is different from OS-PCA compared to smoothed PCA and plain PCA. The description in the methods section gives it inner workings but does not describe in plain English what is happening.

As the manuscript is about introducing OS-PCA as an improved method for analyzing time course data, the conclusion that smoothed PCA and OS PCA show the same results (lines 235-236 and line 249-25) are rather disappointing. What is the added value of OS PCA compared to smoothed PCA?

Line 109: please explain the reason of introducing and the use of the auxiliary variable. What can be used of an auxiliary variable? What are the requirements? Can anything be an auxiliary variable in this context? Does the use of the auxiliary variables makes it supervised?

Minor comments:

Line 77 specify ‘same type of data’

Adding legends in the figures to make them easier to read

Author Response

Responses to the Reviewers

Reviewer 1:

Thank you very much for your kind and useful comments on our manuscript. Our responses to them are given below.

  1. In response to your comment, we added the summary of main feature of each method to understand the differences among PCA, smoothed PCA and OS-PCA in Table1 (page 10-11).
  2. As you point out, the advantage of OS-PCA compared with smoothed PCA is not shown in PC scores. However, the advantage of OS-PCA is that statistical property of loading can be interpreted. As a result, we can select significant metabolites by using statistical hypothesis testing of OS-PC loadings. This point has already been written in abstract, introduction and conclusion.
  3. In response to your comment, we added the description about the reason for introducing and the use of the auxiliary variable (page 8, line 275-277).

“Similarly, auxiliary variable s of OS-PCA is essential to interpret OS-PC loading statistically so that statistical hypothesis testing of OS-PC loadings can be performed, as explained in section 3.4.”

  1. In response to your comment, we modified the description (page2, line 76-78).

“In this study, we describe an orthogonal smoothed PCA (OS-PCA) method that was designed to handle the same type of data that smoothed PCA deals with, for example, samples that were acquired with a time course or rank order.”

  1. In response to your comment, we added the legend for all figures.

Additionally, as requested by the editorial office, the order of the Results and Discussion section and the Methods section has been switched.

Reviewer 2 Report

This manuscript describes the application of orthogonal smoothed principal component analysis to various metabolomics datasets to highlight its benefits over conventional and smoothed PCA approaches. Overall, the manuscript is well-written and provides a novel contribution to metabolomics data analysis. 

My only comment to the authors is the limitation of their reporting to only 1 or 2 principal components. It would be interesting to see how these methods compare in their ability to extract biological or phenotypic information beyond just the first two principal components since it often takes many more components than this to capture the majority of the information. Would it be possible to report, perhaps in the supplement, how these methods compare out to the first 5 PCs? If not, the authors should provide a statement in the text justifying why they chose to limit the focus of their analysis to only 1 or 2 PCs.

Author Response

Thank you very much for your kind and useful comments on our manuscript.

In response to your comment, we added scatter plot of PC3, PC4 and PC5 (or PC6) for dataset1 (Figure S1) and dataset2 (Figure S2) in supplementary figures, and added relevant descriptions to the manuscript (page 3, line 110-115, page 5, line 198-202).

“We also calculated PC3, PC4, and PC5 scores (Figure S1). We confirmed the fluctuation trend against incubation time of X2180 cultured in SD medium without amino acids differed from the trends of BY4742 and X2180 cultured in SD medium with amino acids in PC4 and PC5. However, the contribution ratios of PC4 and PC5 were very small—3.80% and 0.88%, respectively—so we did not consider these PCs in the subsequent analysis.”

“We performed a PCA (Figure 3) and confirmed that the differences among the green tea leaf samples were reflected in PC1 and partial association with the taste ranking was reflected in PC2; however, no clear association with the taste ranking was detected. We also calculated PC3, PC4, and PC5 scores (Figure S2), but their associations with the taste ranking were not confirmed.”

Additionally, as requested by the editorial office, the order of the Results and Discussion section and the Methods section has been switched.